# Evaluation of Hydroxycarboxylic Acid Receptor 1 (HCAR1) as a Building Block for Genetically Encoded Extracellular Lactate Biosensors

**DOI:** 10.3390/bios12030143

**Published:** 2022-02-25

**Authors:** Joel Wellbourne-Wood, Marc Briquet, Maxime Alessandri, Francesca Binda, Maylis Touya, Jean-Yves Chatton

**Affiliations:** 1Department of Fundamental Neurosciences, University of Lausanne, 1005 Lausanne, Switzerland; joel.wellbournewood@gmail.com (J.W.-W.); marc.briquet@unil.ch (M.B.); maxime.alessandri@unil.ch (M.A.); francesca.binda@unil.ch (F.B.); maylis.touya@unil.ch (M.T.); 2Cellular Imaging Facility, University of Lausanne, 1005 Lausanne, Switzerland

**Keywords:** lactate, genetically encoded fluorescent indicator, GPR81, HCAR1, circularly permuted green fluorescent protein

## Abstract

The status of lactate has evolved from being considered a waste product of cellular metabolism to a useful metabolic substrate and, more recently, to a signaling molecule. The fluctuations of lactate levels within biological tissues, in particular in the interstitial space, are crucial to assess with high spatial and temporal resolution, and this is best achieved using cellular imaging approaches. In this study, we evaluated the suitability of the lactate receptor, hydroxycarboxylic acid receptor 1 (HCAR1, formerly named GPR81), as a basis for the development of a genetically encoded fluorescent lactate biosensor. We used a biosensor strategy that was successfully applied to molecules such as dopamine, serotonin, and norepinephrine, based on their respective G-protein-coupled receptors. In this study, a set of intensiometric sensors was constructed and expressed in living cells. They showed selective expression at the plasma membrane and responded to physiological concentrations of lactate. However, these sensors lost the original ability of HCAR1 to selectively respond to lactate versus other related small carboxylic acid molecules. Therefore, while representing a promising building block for a lactate biosensor, HCAR1 was found to be sensitive to perturbations of its structure, affecting its ability to distinguish between related carboxylic molecules.

## 1. Introduction

Since its discovery at the end of the 18th century and for almost two centuries, lactate has been viewed as a waste product of the metabolism generated during hypoxia, and having several adverse effects, such as muscle soreness and fatigue. However, in the 1980s, evidence started to accumulate showing that lactate could in fact be utilized as a useful metabolic fuel fuel [1], i.e., that it could be transferred from glycolytic cells to oxidative cells for energizing respiration. Since then, several studies have indicated the valuable contribution of lactate as a metabolic substrate in tissues, including muscle [2], as a neuroprotective agent, and more recently as a signaling molecule, for reviews, see, e.g., [3]. In the central nervous system, lactate was proposed to play an important role as energy substrate for neurons [4]. Extracellular brain lactate levels are estimated to be in the low millimolar range at resting state [5,6] and to undergo a two-fold increase during synaptic activity [7]. During intense physical exercise, plasma lactate, which can cross the blood–brain barrier, can rise to 10–20 mM [8].

It has recently been found that energy substrates and metabolites of the energy metabolism have extracellular signaling properties by acting through G-protein-coupled receptors (GPCRs) [9,10]. Lactate is the endogenous ligand of one such receptor, originally named GPR81, now known as hydroxycarboxylic acid receptor 1 (HCAR1) [11]. HCAR1 was initially described as markedly expressed in adipocytes, where its activation induces the inhibition of lipolysis through the activation of a G_i_-dependent intracellular pathway [12]. Our research group demonstrated that extracellular L-lactate and non-metabolized HCAR1 agonists decrease the synaptic activity of CNS neurons [13,14,15].

Methods for detecting lactate levels include enzymatic assays, amperometric lactate biosensors, microdialysis, or sniffer cells-based detection [16]. Research on lactate effects in living tissue would critically benefit from a fluorescence imaging approach. Genetically encoded fluorescent indicators have been developed for a long list of analytes including, notably, cations (Ca^2+^, Zn^2+^, H^+^), intracellular metabolites (ATP, GTP, cGMP, glucose, etc.), and extracellular neurotransmitters (glutamate, dopamine, noradrenaline, etc.) [17]. A ratiometric FRET biosensor for lactate, named Laconic, has been developed [18] but is only applicable to intracellular lactate detection [19]. Recently, a genetically encoded biosensor for extracellular lactate was demonstrated that makes use of a prokaryotic lactate periplasmic binding protein TTHA0766 coupled to a circularly permuted GFP (cpGFP) that was addressed to the extracellular side of the plasma membrane [19].

In our study, we investigated whether the strategy used previously for several neurotransmitters based on their respective specific GPCRs [17] could be deployed to develop a lactate-sensitive fluorescence biosensor, and constructed an intensiometric fluorescent lactate-sensitive biosensor based on the HCAR1 lactate receptor coupled to cpGFP.

## 2. Materials and Methods

### 2.1. Engineering of Constructs

The nucleotide sequences of mouse and human HCAR1 were obtained from GenBank (KU285433.1 and KU285432.1, respectively). The gene sequences encoding cpGFP used for the constructs, having LARS as the acronym, were those published previously [20,21]. Constructs were cloned and expressed using the GenScript Biotech (Leiden, Netherlands) or the VectorBuilder (Chicago, IL, USA) platforms. Transmembrane domains of HCAR1 were determined according to the Uniprot database and implied by the published aligned sequences comparing >800 GPCRs [22]. The alignment was also compared to the alignments of other GPCR-based sensors [20]. Linker sequences (N- and C-terminal ends of cpGFP) were those used for other similar biosensors [20,21]. Amino acid sequences used are indicated in Table 1. Predictions of tertiary structures of the LARS constructs were performed using the IntFOLD tool [23] accessed through its integrated protein structure and function prediction server [24]. Three-dimensional visualizations of the constructs were created using Mol* [25], accessed through the RCSB Protein Databank portal [26].

### 2.2. Cell Culture and Transfection

All animal experimentation procedures were carried out in accordance with the recommendations of the Swiss Ordinance on Animal Experimentation and were specifically approved for this study by the Veterinary Affairs of the Canton Vaud, Switzerland (authorizations# VD1288 × 8) and conformed to the ARRIVE guidelines. Cortical astrocytes in primary culture were prepared from 1- to 3-day-old C57BL/6N mice as described previously [27]. Astrocytes were plated on coverslips and cultured for 2 weeks in DMEM (#D7777, Sigma, Schaffhausen, Switzerland) medium plus 10% FCS before experiments. HEK293 were cultured in DMEM plus 10% FCS, and plated on poly-D-lysin (#P6407, Sigma, Schaffhausen, Switzerland) coated coverslips.

Transfection with the plasmids carrying the LARS constructs was performed using Lipofectamine 2000 (#11668019, Invitrogen, Basel, Switzerland) for 5 h. For HEK293 cells, 0.5 µg DNA was applied in Opti-MEM medium (#31985062, Gibco, Thermo Fisher Scientific, Basel, Switzerland), whereas for astrocytes, 1 µg DNA was applied in Neurobasal medium (#21103049, Gibco, Thermo Fisher Scientific, Basel, Switzerland). HEK293 cells were transfected 1 day post-splitting and primary astrocytes were transfected at DIV 14. In some experiments, astrocytes were loaded with the red fluorescent dye sulforhodamine 101 (SR101) applied in the bath at 1µM for 10 min. Cells were used 48 h after transfection, unless otherwise indicated.

### 2.3. Widefield Fluorescence Imaging

Widefield imaging was performed using: (1) An upright epifluorescence microscope (FN1, Nikon, Tokyo, Japan) using a 40 × 0.8 N.A. water-immersion objective lens. Fluorescence was excited using an LED light source (Lambda 421, Sutter Instr., Novato, CA, USA) and detected through a 535 ± 15 nm filter (Chroma, Bellows Falls, VT, USA) using an Evolve EMCCD camera (Photometrics, Tucson, AZ, USA). (2) An epifluorescence inverted microscope (Zeiss Axiovert 100 M) equipped with a high numerical aperture fluorescence objective (40 × 1.3 N.A. oil-immersion), and fast holographic monochromator (Polychrome II, Till Photonics, Planegg, Germany) coupled to a Xenon lamp for fluorescence excitation, with detection using an EMCCD camera (LUCA-R, Andor, Belfast, UK). LARS excitation spectra were obtained by driving the monochromator light source in the range 350–500 nm, with detection of fluorescence through a 535 ± 20 nm filter (Chroma). Digital image acquisition and time series were computer-controlled using the Metafluor software (RRID:SCR_014294). Experimental solutions for live imaging contained (mM): NaCl 160, KCl 5.4, HEPES 20, CaCl_2_ 1.3, MgSO_4_ 0.8, NaH_2_PO_4_ 0.78, glucose 5 (pH 7.4) and were bubbled with air. Solutions containing lactate, pyruvate, β-hydroxybutyrate (BHB), or mannitol had their NaCl concentration correspondingly decreased to ensure iso-osmolarity.

### 2.4. Confocal Imaging

Cells expressing LARS constructs were observed on an LSM780 confocal microscope (Carl Zeiss, Germany) using oil immersion objectives (20×, 40×, or 63× PlanApo). cpGFP was excited at 488 nm and detected in the range 493–569 nm. SR101 was excited at 561 nm and detected in the range 569–712 nm. The GaAsP 32-channel Quasar spectral detector of the LSM780 was used to measure emission spectra of samples using 488 nm fluorescence excitation. ImageJ software (RRID:SCR_003070) was further used for downstream image processing and analysis.

### 2.5. Two-Photon Imaging

Two-photon imaging was carried out using a custom-built two-photon microscope with a 20 × 0.95 N.A. water-dipping objective (Olympus, Tokyo, Japan). Fluorescence excitation was performed using a Chameleon Vision S femtosecond infrared laser including group velocity dispersion compensation (Coherent, CA, USA). Fluorescence emission was measured at 525 ± 25 nm (Semrock FF01–525/50). To measure two-photon excitation spectra, the infrared laser intensity was adjusted to yield a constant value throughout the Ti:Sapphire emission spectrum using an acousto-optic modulator (AA Opto-Electronic, Orsay, France). Image acquisition was performed using custom software in the Labview (RRID:SCR_014325) environment.

### 2.6. Data Analysis

Bar graphs show mean ± SEM based on quantification of individual traces, of which examples are shown. The mean fluorescence intensity in regions of interest selected around 5–6 cells (later referred to as cell groups) was measured and quantified in the image series. Graphs and nonlinear curve fitting for EC_50_ calculations were carried out using KaleidaGraph software (Synergy Software; RRID:SCR_014980).

### 2.7. Chemicals and Drugs

Unless specified, chemicals were from Sigma-Aldrich.

## 3. Results and Discussion

### 3.1. Design and Characterization of HCAR1-Based Lactate Sensors

A major class of genetically encoded intensiometric fluorescent sensors is based on engineered chimeric seven transmembrane (7 TM) GPCR that are integral membrane proteins. The binding of the ligand to the extracellular receptor domain causes conformational changes that are transmitted to an integrated fluorescent protein, modulating its fluorescent properties. Several biosensors for neurotransmitters have cpGFP integrated on the intracellular side of the receptor protein in the third intracellular (IC3) loop [17]. We based our design on successful biosensors for dopamine, norepinephrine, melatonin [20], and for 5-hydroxytryptamine (5-HT, serotonin) [21]. The design strategies for main candidates are summarized in Table 1.

The first listed constructs (LARS1.1 and LARS1.2) did not yield any or only sparse intracellular fluorescence, possibly because of the too close proximity of cpGFP to the plasma membrane and intracellular segments of HCAR1, which prevented it from properly folding. Thus, longer stretches of the IC3 loop were kept, along with different linkers. In addition, two different membrane targeting sequences were compared: the hemagglutinin secretion motif (HA) and the immunoglobulin κ-chain leader sequence (IgK). IgK proved to significantly better address the protein constructs to the plasma membrane. The complete sequence of best candidate LARS1.8 is presented in Appendix A.

The three optimized sensors—LARS1.5, LARS1.8, and LARS1.10—yielded green fluorescence associated with the plasma membrane when transfected in HEK293 cells (Figure 1).

Compared to LARS1.5, LARS 1.8 and LARS1.10 were more clearly localized to the plasma membrane. Primary mouse astrocytes transfected with the LARS1.8 plasmid displayed plasma membrane expression that extended to their finest lamellipodia (Figure 2), which strongly contrasted with the distribution of the red cytosolic dye SR101 that was only visible in the thickest regions of these flat cultured astrocytes. Note that a profile plot, as performed on HEK293 cells, would not adequately reflect membrane localization in those extremely thin cells.

In order to further support the correct folding and localization of the protein constructs, we conducted molecular modeling simulations using the IntFold [23]. Figure 3 graphically depicts the LARS1.8 construct domains and shows the predicted tertiary structure obtained for the LARS1.8 amino acid sequence. The expected transmembrane domains of the receptor part as well as cpGFP tertiary structure appear correctly folded. Similar successful predicted 3D structures were obtained for LARS1.5 and LARS1.10, and provided the indication that the constructs would be correctly inserted in the plasma membrane.

### 3.2. Biosensor Sensitivity to Lactate

The fluorescence emission and excitation spectra of LARS1.8 recorded on live HEK293 cells are depicted in Figure 4a,b. The excitation spectrum displayed the typical −400 nm and −490 bands of cpGFP corresponding to the neutral and anionic form of the protein, while emission peaked at −515 nm as observed with the calcium indicator series GCaMP [28]. Two-photon excitation of LARS1.8 using a pulsed femtosecond infrared laser is depicted in Figure 4c. The optimal two-photon excitation wavelength was found at 950 nm, while a second two-photon absorption band was observed at −1000 nm.

LARS 1.8 was first selected for functional testing as it displayed the most clearly defined membrane expression. The fluorescence of LARS1.8 expressed in HEK293 cells was found to exhibit pronounced changes following lactate application. Interestingly, an inverse response was observed corresponding to rapid and reversible fluorescence decreases (Figure 5a). This inverse response could already be seen in the excitation spectra measured in the presence and absence of lactate (Figure 4b), which indicated that lactate caused a homogenous decrease in LARS1.8 fluorescence. The dynamic range of fluorescence over the binding curve is in the range of 30%, providing an ample fluorescence window for sensing concentration changes. Similar inverse responses to lactate application were also observed with the LARS1.5 and LARS1.10 constructs expressed in HEK293 cells (Figure 5b). The apparent affinity of the three LARS constructs was estimated to be 4.5 ± 1.5 mM, 1.5 ± 1.5 mM, and 5.6 ± 2.1 mM for LARS1.5, LARS1.8, and LARS1.10, respectively. These apparent EC_50_ values affinities are in the range reported for HCAR1 [12,13].

### 3.3. Biosensor Selectivity

In the next phase, we investigated whether the LARS constructs were able to discriminate lactate from related molecules. Figure 6 shows that LARS1.8 displayed similar responses to L-lactate, pyruvate, BHB, and D-lactate when applied at the same concentration (10 mM). The LARS1.5 construct, which differs from LARS1.8 by the membrane addressing sequence, also responded to pyruvate and BHB. As the original HCAR1 protein, from which LARS constructs are derived, has selectivity for lactate over these other molecules, one can hypothesize that the addition of cpGFP with its 11 amino acid long linkers in the third intracellular loop is perturbing the receptor such that it loses its selectivity for lactate over other hydroxycarboxylic acids. We thus tested whether LARS1.10, a construct having shorter linkers (5 amino acids) flanking cpGFP, would exhibit a different behavior. Figure 6 shows that LARS1.10 was also not able to distinguish lactate from pyruvate or BHB when applied at 10 mM. Importantly, we show that 10 mM mannitol, a molecule of similar molecular size but that is not a hydroxycarboxylic acid, did not cause significant fluorescence responses, indicating that the LARS constructs have selectivity for hydroxycarboxylic acids.

Thus, it appears that inserting the cpGFP fluorescent protein in the third intracellular loop, as was successfully performed for several other biosensors based on GPCRs (e.g., for dopamine, norepinephrine, serotonin, etc.), had the effect of perturbing the selectivity for lactate found in the original HCAR1 protein. We postulate that the linkers flanking cpGFP were not the only culprits, as the same result was found for two linkers that differed both in their length and in their amino acid composition.

## 4. Conclusions and Outlook

Overall, in this study, we describe—to our knowledge—the first attempt at using the lactate receptor HCAR1, a G_i_-coupled GPCR, to build a fluorescent biosensor for extracellular lactate. We identified that addressing the protein constructs to the plasma membrane was very efficiently achieved using the IgK leading sequence. The strategy appears promising as the candidate sensors respond to lactate at physiologically relevant concentrations with cellular resolution. Rapid fluorescence changes were observed with kinetics that are compatible with the expected kinetics of physiological lactate variations found in tissues, for instance in response to neuronal activity or to intense physical exercise. Nevertheless, further improvements of the sensors need to address the issue of selectivity. When screening for improved versions of HCAR1-based lactate sensors, as was performed for other sensors such as the dopamine or serotonin sensors, one needs to include the double condition of maximum fluorescence change induced by lactate coupled with the lack of response to other related molecules, such as pyruvate or BHB. Ideally, variants showing fluorescence increases rather than decreases (inverse response) to lactate application would be advantageous—but not mandatory—for experiments. Compared with the recently published lactate biosensor [19] based on a bacterial extracellular lactate binding protein coupled to GFP, HCAR1-based sensors would arguably have the advantage of positioning the fluorescent protein inside the cell, protecting it from ion fluctuations that may occur in the interstitial milieu in particular in pathological situations. In addition, a biosensor based on HCAR1 that would retain the properties of the original HCAR1 protein could represent a powerful tool for the screening of new agonists or antagonist of the receptor, as well as for allosteric modulators. In conclusion, imaging approaches of lactate biosensors offer the prospect of precisely analyzing lactate concentration fluxes within living tissues, including in microdomains around cells, which will bring invaluable information for both physiological and pathological conditions.

## Figures and Tables

**Figure 1 biosensors-12-00143-f001:**
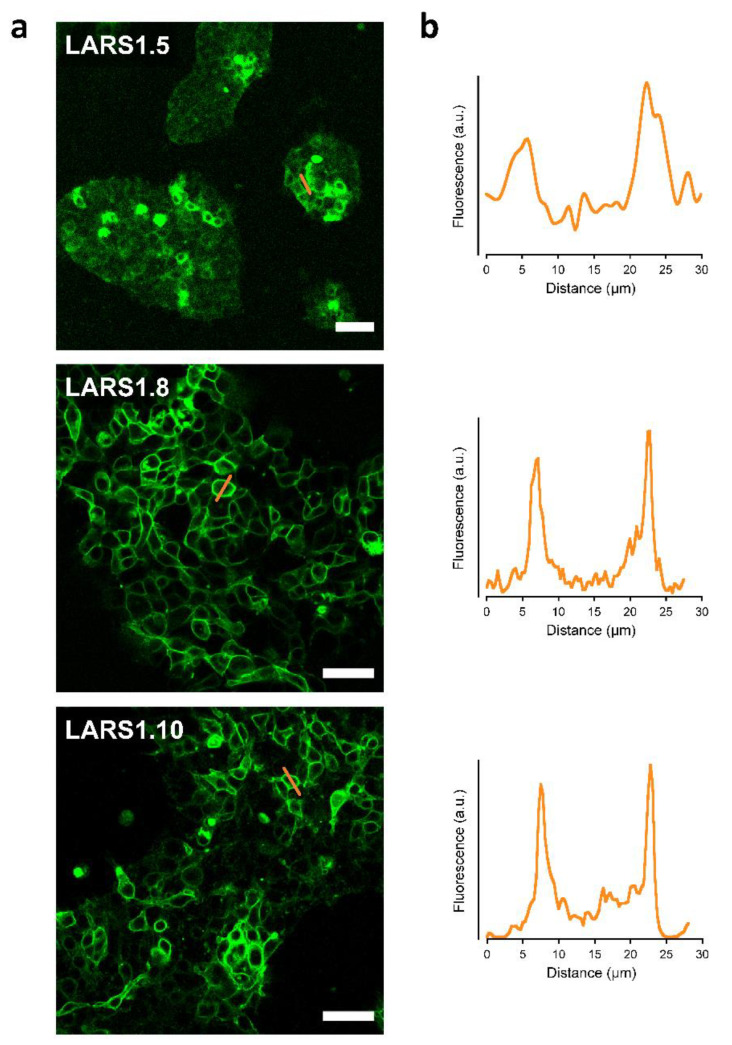
Cellular expression of candidate biosensors in HEK293 cells. Confocal images of live HEK293 cells transfected with LARS1.5, LARS1.8, and LARS1.10 and observed 48 h after transfection. Images show endogenous cpGFP fluorescence (λ_ex_ = 488 nm) (**a**). A representative profile plot across cells (at sites indicated by orange lines in left panel) is shown next to the candidate biosensors (**b**). Scale bar, 50 µm.

**Figure 2 biosensors-12-00143-f002:**
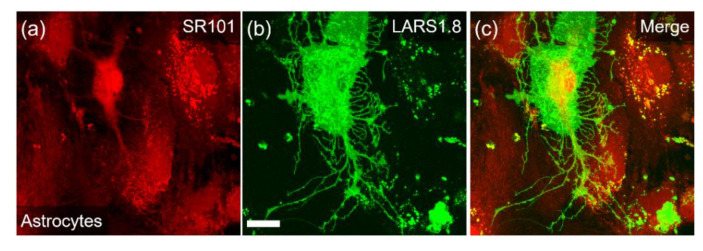
Cellular expression in primary mouse astrocytes. Confocal images of live mouse astrocytes transfected with LARS1.8 and observed 48 h after transfection. (**a**) SR101 dye staining (λ_ex_ = 561 nm), (**b**) endogenous cpGFP fluorescence (λ_ex_ = 488 nm), and (**c**) merged image. Note the cpGFP green staining extending to the finest membrane processes. Scale bar, 20 µm.

**Figure 3 biosensors-12-00143-f003:**
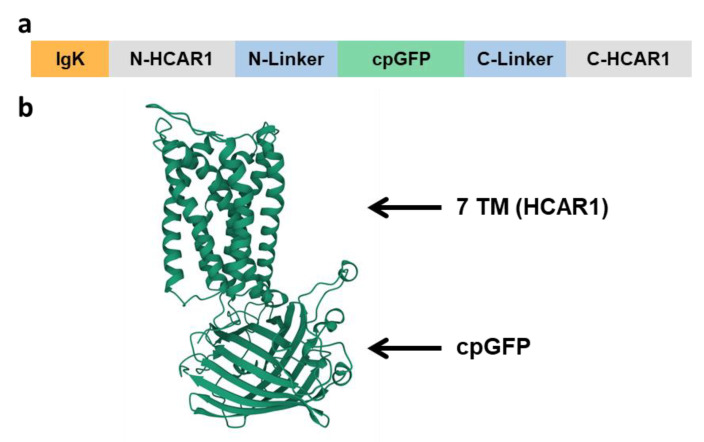
Design of candidate biosensors and predicted tertiary structure of the protein. (**a**) The general design of the constructs comprises a plasma membrane addressing sequence (IgK or HA) fused at the N-terminal side of the protein. cpGFP was inserted in the intracellular loop 3 and flanked with linker sequences at the N- and C-terminal side of the loop. (**b**) Tertiary structure prediction of LARS1.8 highlighting the seven-transmembrane domains (7 TM) of HCAR1 and the cpGFP insertion. For calculations of structure prediction, the disordered N- and C-termini of the sequence were trimmed to keep 15 amino acid residues before the start of transmembrane domain 1 and 10 residues after transmembrane domain 7 on N-terminal side. The structure prediction yielded an overall high level of confidence (*p* = 0.0024).

**Figure 4 biosensors-12-00143-f004:**
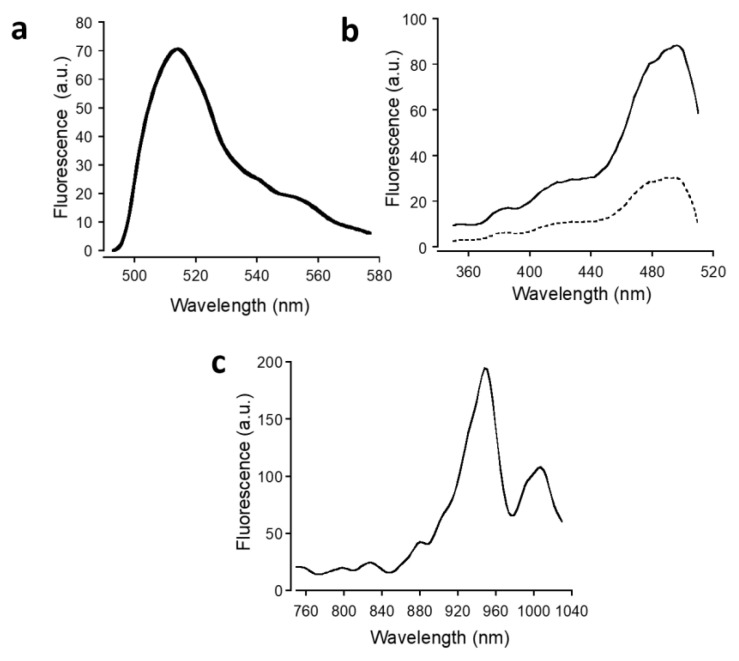
Fluorescence spectral properties. (**a**) In situ fluorescence emission spectrum of LARS1.8 (λ_ex_ = 488 nm) in live HEK293 cells. (**b**) One-photon excitation spectra in the absence (plain line) and in the presence of lactate (10 mM, dotted line). (**c**) Two-photon excitation spectrum.

**Figure 5 biosensors-12-00143-f005:**
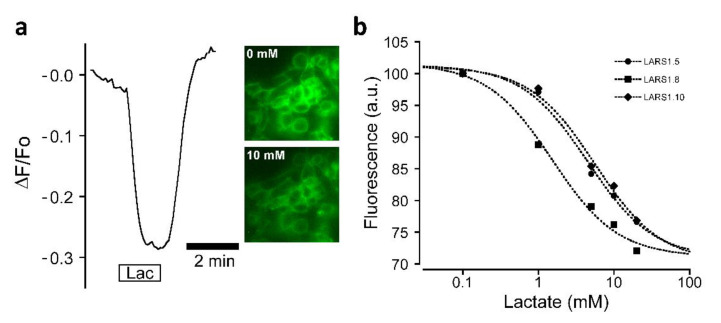
Responses of candidate biosensors to lactate application. (**a**) Representative fluorescent curve of LARS1.8 response to 10 mM lactate application. Widefield fluorescence images of recorded a group of HEK293 cells before and during application of lactate are shown (λ_ex_ = 490 nm). (**b**) Lactate concentration dependence of LARS1.5, LAR1.8, and LARS1.10 fluorescence responses. Data points are mean values measured 6–7 groups of cells for each construct. EC_50_ values were obtained by nonlinear curve fitting using the Levenberg–Marquardt algorithm. Data are normalized to the fluorescence measured in the absence of lactate.

**Figure 6 biosensors-12-00143-f006:**
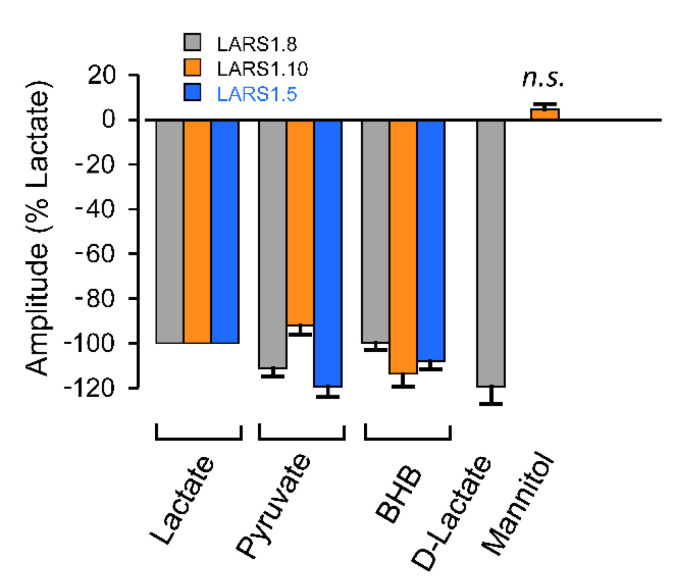
Selectivity candidate biosensor responses. The amplitude of responses of LARS1.8 (*n* = 16 cell groups), LARS1.10 (*n* = 14–17 cell groups) and LARS1.5 (*n* = 3 cell groups) to pyruvate (10 mM), and β-hydroxybutyrate (10 mM). Additionally, D-lactate (10 mM), sensitivity and lack of sensitivity to mannitol (10 mM) were observed. Responses were normalized to the response to lactate (10 mM) on the same cells.

**Table 1 biosensors-12-00143-t001:** Summary of main construct design rationale.

Construct Name	Membrane Targeting Sequence	Linkers	Description	Results/Observations
LARS1.1	HA secretory sequence	Based on dLight 1.1 or 1.2 ^1^ N-linker: LSSLI C-linker: NHDQL	Use mouse HCAR1 gene; replace entire IC3 loop	No measurable fluorescence
LARS 1.2	HA secretory sequence	Based on dLight 1.1 or 1.2 ^1^ N-linker: LSSLI C-linker: NHDQL	Use human HCAR1 gene; replace entire IC3 loop	Some fluorescence, intracellular localization, lysosomes or ER
LARS 1.3	HA secretory sequence	Based on dLight 1.1 or 1.2 ^1^ N-linker: LSSLI C-linker: NHDQL	Using mouse HCAR1 gene; replace part of IC3 loop	Weak fluorescence, intracellular localization, lysosomes or ER
LARS 1.5 *	HA secretory sequence	Optimized linkers for B2AR and MT2R ^1^ N-linker: QLQKIDLSSLI C-linker: NHDQDIKQLQ	Use mouse HCAR1 gene; replace part of IC3 loop	Fluorescence partly intracellular and plasma membrane in several cells
LARS 1.7	HA secretory sequence	Based on dLight 1.1 or 1.2 ^1^ N-linker: LSSLI C-linker: NHDQL	Use human HCAR1 gene; replace entire IC2 loop	Weak and sparse fluorescence, intracellular localization
LARS 1.8 *	IgK secretory sequence	Optimized linkers for B2A and MT2 receptors ^1^ N-linker: QLQKIDLSSLI C-linker: NHDQDIKQLQ	Use mouse HCAR1 gene; replace part of IC3 loop; use IgK secretory sequence and mutated cpGFP of GRAB_5-HT_ ^2^	Robust plasma membrane fluorescence
LARS 1.10 *	IgK secretory sequence	Based on GRAB_5HT_ ^2^ N-linker: MFLNG C-linker: GFATA	Use mouse HCAR1 gene; replace part of IC3 loop; use IgK secretory sequence; use GRAB_5-HT_ linkers and mutated cpGFP ^2^	Robust fluorescence with mixed intracellular and membrane localization

***** Kept for further testing. ^1^ dLight is a dopamine biosensor [20]. ^2^ GRAB_5HT_ is a serotonin biosensor [21]. B2A, β2-adrenergic receptor; MT2R, melatonin type 2 receptor; HA, hemagglutinin leader sequence; IgK, immunoglobulin kappa light chain leader sequence.

## Data Availability

Not applicable.

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
