# Peer review of "Evaluation of Hydroxycarboxylic Acid Receptor 1 (HCAR1) as a Building Block for Genetically Encoded Extracellular Lactate Biosensors"

_biosensors, 2022, doi:10.3390/bios12030143_

Round 1
Reviewer 1 Report
Authors showed a genetically encoded sensor for lactate detection in cells. Although other research group have already reported genetically encoded lactate sensors recently and the sensor in this work is not as good as the reported one, I think this work is meaningful because of usage of hydroxycarboxylic acid receptor 1 (HCAR1) as sensing domain of the sensor. The sensor construct using different sensing domain provides a new basic construct for genetically encoded lactate sensor and will become a fundament to develop better genetically encoded lactate sensor. Its publication will help other researchers to better design and construct genetically encoded lactate sensor. Thus, I would like to publish this work in biosensor.
I have some question about this work.
- I found authors would like to use confocal microscope to observe fluorescence of the sensor. Why? Is the sensor dim in fluorescence? What is brightness of the sensor when compared to GFP? 1% brightness? 10% brightness or other number? Brigtness is very important for application of sensor.
- Authors had linker optimization for the sensor. What is detail? How many mutants? What is performance of the sensor with different amino acid linkers?
- Authors performed all test inside cells. Did authors test the sensor by purified proteins? If yes, how about result?
Author Response
Manuscript ID: biosensors-1613352
Evaluation of hydroxycarboxylic acid receptor 1 (HCAR1) as a building block for genetically encoded extracellular lactate biosensors.
Authors: Joel Wellbourne-Wood , Marc Briquet , Maxime Alessandri , Francesca Binda , Maylis Touya , Jean-Yves Chatton
The authors would like to thank both reviewers for their positive appreciation of our work.
Answers to reviewer#1:
1-Brightness of the fluorescent sensor:
- We agree that sensor brightness is of critical importance. Overall, we found that the sensors tested (LARS1.5, LARS1.8, LARS1.10) provided a bright enough fluorescence.
- Use of confocal microscopes: we used confocal microscopes only to obtain more detailed images and to record fluorescence emission spectra of the biosensors. All the live-cell imaging experiments (fig. 5 and 6) were performed on widefield fluorescence microscopes running at low light level of excitation.
- Brightness comparison with GFP: it is difficult to compare with plain GFP as the staining of our biosensor is restricted to the plasma membrane and therefore corresponds to a very low volume of fluorescent protein. Moreover, the fluorescence depends on the concentration of the measured analyte. Nevertheless, for constructs LARS1.8 and LARS1.10 (the two most efficient ones of the series), we used the optimized cpGFP sequence reported to be brighter by Wan et al. Nat Neurosci 2021.
2-Linkers:
- We added the amino acid linker sequences used in Table 1 and added an explanatory sentence in the methods section.
3- Testing in purified proteins:
- The LARS biosensors are based on GPCRs that are integral membrane proteins. Such membrane proteins are not soluble. If purified, one would expect that the sensor protein would only show GFP fluorescence without sensitivity to analytes. We therefore decided not to attempt purifying them.
Reviewer 2 Report
The authors of this paper developed novel fluorescent biosensors for the detection of lactate in living tissues. The sensors are based on the HCAR1 lactate receptor coupled to cpGFP and they show fluorescence with intracellular or/and plasma localization. The results are clearly presented and critically evaluated. The work may inspire further research leading to highly efficient fluorescent lactate sensors. The manuscript is written thoroughly, I recommend it for publication without change in Biosensors.
Author Response
We thank the reviewer for his positive appreciation of our work.
JY Chatton